# Dissecting Microbiome-Derived SCFAs in Prostate Cancer: Analyzing Gut Microbiota, Racial Disparities, and Epigenetic Mechanisms

**DOI:** 10.3390/cancers15164086

**Published:** 2023-08-14

**Authors:** Thabiso Victor Miya, Rahaba Marima, Botle Precious Damane, Elisa Marie Ledet, Zodwa Dlamini

**Affiliations:** 1SAMRC Precision Oncology Research Unit (PORU), DSI/NRF SARChI Chair in Precision Oncology and Cancer Prevention (POCP), Pan African Cancer Research Institute (PACRI), University of Pretoria, Pretoria 0028, South Africa; 2Department of Surgery, Level 7, Bridge E, Steve Biko Academic Hospital, Faculty of Health Sciences, University of Pretoria, Pretoria 0007, South Africa; 3Tulane Cancer Center, Tulane Medical School, New Orleans, LA 70112, USA

**Keywords:** prostate cancer, human microbiome, microbiota, short-chain fatty acids, metabolites, racial disparities, epigenetic modifications

## Abstract

**Simple Summary:**

Prostate cancer (PCa) is still the most diagnosed cancer and the second cause of death among men worldwide. Studies have shown that the human microbiome and its metabolites such as short-chain fatty acids (SCFAs) play a key role in PCa development and progress, as well as response to anticancer treatments. Furthermore, studies have reported racial disparities in terms of microbiome composition and SCFA content across different human cancers, including colon, cervical, and colorectal cancer. Lastly, studies have shown that epigenetic modifications also play a crucial role in carcinogenesis and response to various treatment interventions. Therefore, more research is needed to fully understand underlying molecular mechanisms that lead to PCa in terms of the human microbiome, microbiome-derived metabolites, race, and epigenetic modifications. Importantly, this will advance efforts into personalized treatment strategies across various cancers, including PCa.

**Abstract:**

Prostate cancer (PCa) continues to be the most diagnosed cancer and the second primary cause of fatalities in men globally. There is an abundance of scientific evidence suggesting that the human microbiome, together with its metabolites, plays a crucial role in carcinogenesis and has a significant impact on the efficacy of anticancer interventions in solid and hematological cancers. These anticancer interventions include chemotherapy, immune checkpoint inhibitors, and targeted therapies. Furthermore, the microbiome can influence systemic and local immune responses using numerous metabolites such as short-chain fatty acids (SCFAs). Despite the lack of scientific data in terms of the role of SCFAs in PCa pathogenesis, recent studies show that SCFAs have a profound impact on PCa progression. Several studies have reported racial/ethnic disparities in terms of bacterial content in the gut microbiome and SCFA composition. These studies explored microbiome and SCFA racial/ethnic disparities in cancers such as colorectal, colon, cervical, breast, and endometrial cancer. Notably, there are currently no published studies exploring microbiome/SCFA composition racial disparities and their role in PCa carcinogenesis. This review discusses the potential role of the microbiome in PCa development and progression. The involvement of microbiome-derived SCFAs in facilitating PCa carcinogenesis and their effect on PCa therapeutic response, particularly immunotherapy, are discussed. Racial/ethnic differences in microbiome composition and SCFA content in various cancers are also discussed. Lastly, the effects of SCFAs on PCa progression via epigenetic modifications is also discussed.

## 1. Introduction

Prostate cancer (PCa) is the second most diagnosed cancer among men worldwide, after lung cancer [1]. In addition, PCa is also the second most common cause of cancer-induced fatalities, globally. In 2020, PCa was estimated to account for a total of 1,414,259 new cases and 375,304 related fatalities worldwide [2]. Although there is currently no cure for advanced PCa, the five-year survival rate of localized PCa is >99%, [3] [4]. Advanced PCa is defined by recurrence post definitive local therapy such as radiation and/or surgery, or with evidence of metastases at any point [5]. Conversely, metastatic PCa has a 31% five-year survival rate [4] and thus, there is an imperative need for effective novel therapeutic agents against this non-curable disease [5]. However, management of PCa is rapidly changing due to advancements in understanding of its evolution, mutational landscape, and signaling pathways, as well as its resistance mechanisms. In recent years, intense research has focused on the indirect or direct link between cancer and particular microflora of various cancers, PCa included [6]. This research has discovered that the human microbiome plays a crucial role in terms of cancer pathogenesis as well as response to anticancer therapies [7]. This occurs in two main ways: (1) by direct effect on tumors and (2) by indirect induction of metabolic changes, epithelial damages, and immune modulations (Figure 1) [7]. Furthermore, it was reported that certain microflora can delay tumor growth [8]. Thus, it is crucial to establish the effects of the human microbiome on cancer to be able to formulate novel therapeutic treatments [9,10]. This review discusses the role of microbiome-derived short-chain fatty acids in PCa carcinogenesis and how these metabolites impact therapeutic response with more focus on cancer immunotherapy. The involvement of microbiome-derived short-chain fatty acids (SCFAs) in facilitating PCa carcinogenesis and their effect on PCa therapeutic response is discussed, particularly immunotherapy. Racial/ethnic differences in microbiome composition and SCFA content in various cancers are also discussed. Lastly, the effects of SCFAs on PCa progression via epigenetic modifications is also discussed.

### Association between the Human Microbiome and PCa Pathogenesis

The human microbiome has previously been reported to be involved in immunity, neurological and cognitive functions, inflammation, metabolism, and hematopoiesis [11,12]. The human microbiome composition varies depending on the following: genetic factors, type of birth delivery, colonization at birth, lifestyle of the host, dietary factors, diseases, and exposure to antibiotics or other drugs [13,14,15,16,17]. Furthermore, microbiome composition varies based on the environment in which it is located [18]. For example, the gastrointestinal microbiome comprises five bacterial phyla, namely *Proteobacteria*, *Firmicutes*, *Verrucomicrobia*, *Actinobacteria*, and *Bacteroidetes* [19,20,21]. Importantly, differences in the microbiota in relation to their specific location can assist in PCa-targeted therapy. The most predominant anaerobes are *Peptostreptococci*, *Bifidobacteria*, *Ruminococci*, *Eubacteria*, and *Bacteroides* [20]. On the other hand, using PCR and 16S ribosomal ribonucleic acid (RNA) sequencing, the urinary tract has been reported to contain unique microbiome species [22,23,24]. Moreover, the composition of the microbiome within the urogenital tract varies according to gender due to hormonal differences and the physiological differences between age and gender [24]. The female urogenital microbiome mainly comprises *Gardnerella* and *Lactobacillus* genera [25,26]. Conversely, the male urogenital microbiome mainly comprises *Streptococcus*, *Corynebacterium*, and *Staphylococcus* [25,26].

## 2. The “Prostate Microbiome”

### 2.1. Intraprostatic Microbiome

Several research studies have identified microbial composition in PCa tissues. However, it is still unclear if the “prostate microbiome” is unique [27,28]. The challenge with these studies is contamination that ends up giving false positive results [29]. Studies that analyzed the presence of “prostate microbiome” have thus far reported that the prostate microflora composition is the same as in the urethra [30,31]. Hochreister and colleagues carried out 16S rRNA PCR to ascertain the presence or absence of bacterial species in prostate tissues collected from 9 patients and 18 healthy controls. The study detected a significant presence of bacterial deoxyribonucleic acid (DNA) in PCa samples compared to healthy controls. However, the researchers did not identify which bacterial species the detected DNA belonged to [32]. Sfanos et al. [33] caried out 16S ribosomal DNA (rDNA) sequencing on radical prostatectomy tissue core specimens. Notably, bacterial DNA was observed in the prostate tissues. However, when the results were contrasted to the core samples, the specimens were negative. Additionally, there was no significant link observed between bacterial species presence and chronic or acute inflammation. Interestingly, focal regions showed numerous bacterial species that are usually found in urinary-tract infections, these included *Pseudomonas*, *Enterococci*, and *Escherichia* spp. Furthermore, the authors observed other species such as *Actinomyces*, *Acinetobacter*, and *Streptococcus* spp. which are frequently present in the urethral flora in physiological settings. Based on these findings, the authors concluded that the prostate microbiome might not exist and that their findings were merely due to remnant bacterial DNA which was ‘fossilized’ in the prostate [33]. In 2017, another study was conducted in which 16S rDNA next-generation sequencing (NGS) and RNA sequencing were carried out on specimens from 20 patients with an aggressive PCa [27]. The aim of this study was to determine histopathologically the existence of infectious agents in aggressive PCa cases. The researchers reported the existence of the *Enterobacteriaceae* family, of which *Propionibacterium acnes* (*P. acnes*) and *Escherichia* were the most predominant species [27]. All these studies have paved the way for identification of the microbiomes and research into their potential role during PCa pathogenesis [29].

Cavarretta et al. [30] observed that *P. acnes* was the most common bacterial species in PCa patients, and that it was evenly present in all tissues. The role of *P. acnes* in the pro-inflammatory pathway within prostate tissue was confirmed using murine models. These results suggest that *P. acnes* may play a central role in PCa carcinogenesis [34,35]. Additionally, the authors also observed a larger amount of *Staphylococcaceae* in malignant tissues and also a larger proportion of *Streptococcaceae* in non-malignant tissues [30]. It has been speculated that the existence of *Streptococcus* in non-malignant tissues might be a sign of normal microbiome of a healthy prostate tissue [29]. However, it is critically important to note that *Staphylococcus* and *Streptococcus* spp. are among the most common human skin bacteria. Therefore, they represent contamination during laboratory analysis [36]. Feng and colleagues carried out the analysis of tissue specimens from a cohort of 65 patients who underwent radical prostatectomy [28]. In this study, *Escherichia*, *Cutibacterium*, *Pseudomonas*, and *Acinetobacter* were reported to be the most predominant bacterial species in the examined tissues. In addition, the authors did not observe any difference in the adjacent benign tissues [28].

In another study, the presence of pathogens in tissue specimens from a cohort comprising 50 patients who underwent radical prostatectomy and a cohort of 15 benign prostatic hyperplasia (BPH) patients who had undergone prostate transurethral resection procedure were evaluated [37]. This was conducted using pan-pathogen microarray metagenomics analysis (PathoChip). The authors reported a distinct pathogenic microbiome in the specimens from PCa patients. This pathogenic microbiome comprised *Firmicutes*, *Bacteroides*, *Actinobacteria*, and *Proteobacteria* phyla. Furthermore, there was no difference between the microbiota signatures of PCa patients’ samples and samples from those without PCa. Nonetheless, the most important observation was the discovery of *Helicobacter pylori* in >90% of PCa samples, which further confirmed the integration of *H. pylori*-cytotoxin-associated gene A (CagA) into the DNA of the prostate tumor [37]. The CagA gene has been reported as a virulence factor for *H. pylori* and has been linked to gastric cancer pathogenesis via suppression of tumor suppressor genes (TSGs) and actuating of proto-oncogenes [38]. The authors also reported the existence of numerous oncogenic viruses such as human papilloma virus (HPV) 16, HPV 18, and human cytomegalovirus which accounted for 41% of all isolated viruses [37]. In the same year, Miyake et al. [39] assessed the existence of pathogens involved in sexually transmitted infections (STIs) and their role in PCa carcinogenesis. The authors collected samples from 45 PCa and 33 BPH patients and analyzed them for the presence of numerous pathogens. The analyzed pathogens included *Ureaplasma urealyticum*, *Mycoplasma genitalium*, *Chlamydia trachomatis*, *Mycoplasma hyorhinis*, *Neisseria gonorrhoeae,* and HPV 18/16. The authors reported that *Mycoplasma genitalium* was the only species linked with a higher Gleason score and PCa pathogenesis [39]. In 2009, a meta-analysis involving case-control research studies from patients with PCa and healthy controls (HCs) was carried out [40]. The authors reported a significant link between cancer risk and infection history of STI such as HPV and *Mycoplasma genitalium* [40].

### 2.2. Genitourinary Microbiome

For a long time, the urinary tract was thought of as a sterile organ [22]. However, numerous research studies have recently reported the existence of a urinary microbiome that is different from the gut microbiome [24,41,42]. This urinary-tract microbiome signifies the effect of the microbiome in PCa [22]. Since the urinary tract is in close proximity to the prostate gland, it can contaminate it and thus, urinary microbial research studies are crucial in the identification of prostate diseases [7,43]. Numerous studies have isolated different microbial strains from the urine of adult males. These microbes include *Staphylococcus*, *Prevotella*, *Finegoldia*, *Streptococcus*, *Veillonella*, *Propionibacterium*, and *Corynebacterium* [44,45,46]. *P. acnes* can be described as a proinflammatory bacterium that is commonly isolated from the urine of males. The link between *P. acnes*, human PCa, and prostatitis in animal models has previously been reported [26,33,34,47,48,49,50]. Chronic prostatitis is commonly induced by uropathogenic strains of *Enterococci* and *E. coli* [51]. On the other hand, prostatitis induced by *P. acne* and *E. coli* strains can result in hyperplasia and morphological changes [51]. Furthermore, these changes have also been implicated in decreasing a tumor suppressor called NKX 3.1 in the prostate [34]. Two clinical studies have reported that proinflammatory bacteria such as *Propionimicrobium lymphophilum*, *Anaerococcus lactolyticus*, *Streptococcus anginosus*, and *Varibaculum cambriense* are frequent in patients with cancer [52,53]. These results suggest that pro-inflammatory bacteria can potentially induce inflammation on the prostate gland for PCa pathogenesis [54].

Several studies have reported that the human urinary microbiome differs according to gender, disease, and age. However, these studies were different with regard to methodology and sample collection method, as well as inclusion criteria [24,42,46]. For instance, it was previously reported that the genera *Streptococcus*, *Staphylococcus*, and *Corynebacterium* are mainly found in the male urinary microbiome [24,26]. Dysbiosis of the urinary microbiome can be due to factors such as urinary incontinence, puberty, or antimicrobial agents of prostatic secretions, as well as sexual behavior [55,56]. Dysbiosis plays a central role in terms of reaction to urinary-tract diseases, and also affects immune molecules [57,58]. In addition, urinary microbiome variation between individuals plays a significant role in the susceptibility to STIs such as *N. gonorrhoeae* and *C. trachomatis* [44]. Clinical studies have also shown increased levels of PSA associated with STIs, which could indicate involvement of the prostate [59,60]. Additionally, the PCa pathogenesis risk can be exacerbated by a history of inflammatory STIs [61]. Association between male urinary microbiome and prostate diseases such as PCa, BPH, and prostatitis has been reported, and has been discussed elsewhere [26,31,62]. However, more clinical studies that focus on the effects of the urinary microbiome composition on PCa carcinogenesis need to be carried out [63].

### 2.3. Gut Microbiome

Numerous studies have established a potential link between gut microbiome and PCa development and resistance to anticancer therapies [64]. Liss et al. [65] analyzed gut microbiome from 133 USA men who had undergone prostate biopsy. Analysis were carried out on rectal swabs using 16S rRNA sequencing. The authors reported that *Streptococcus* and *Bacteroides* spp. were higher in PCa patients. They then carried out metagenome analysis which showed that the gut-microbiome arginine and folate pathways were significantly modified. Subsequently, the authors suggested that the risk of PCa may potentially be impacted by the gut bacteria [65]. Golombos [66] analyzed the gut microbiota from a cohort of 20 men. Of the 20 men, 12 had high-risk PCa, while 8 had benign prostate hypertrophy [66]. Results demonstrated that the men with PCa had elevated *Bacteroides massiliensis*. Nevertheless, the specific mechanism of action is not yet fully understood [66].

Matsushita and colleagues analyzed the gut microbiome from a cohort of 152 Japanese men who had undergone prostate biopsy [67]. The analysis showed that *Alistipes*, *Rikenellaceae*, and *Lachnospira* were highly elevated in men with increased Gleason PCa. The gut microbiota profile, consisting of 18 gut bacteria, could be used to predict PCa with a higher Gleason score compared to the prostate-specific antigen (PSA) test. In this study, bacterial taxa within high-risk PCa were not impacted by metastasis. These observations suggest that modifications in the gut microbiome in high-risk PCa induces PCa and are not the result of PCa [67].

## 3. Microbiome-Derived Short-Chain Fatty Acids (SCFAs)

It is widely known that cancer is induced by the interaction between host genetics and environmental factors. However, several research studies have highlighted the crucial role that microorganisms play in carcinogenesis [68]. Many carcinogenic microbes such as *Helicobacter pylori*, hepatitis B (HBV) and C (HCV), and HPV have been identified in 20% of all malignancies [69]. An additional group of oncogenic microbes, called the microbiota, were recently reported as key factors in carcinogenesis [68]. Recent years have seen the development of cutting-edge technologies to analyze and quantify human microbiota and link their role in carcinogenesis [70]. Despite these ongoing investigations, the exact role of the microbiota in carcinogenesis is yet to be fully understood [71]. However, studies have shown that bacterial-derived metabolites are the central link between gut microbiota and cancer development [72]. Gut microbiota converts fermentable and non-digestible carbohydrates such as dietary fiber into several SCFAs (Figure 2) [73]. Predominant SCFA members are acetate, butyrate, and propionate (Figure 2). The total intestinal concentration of these SCFAs may reach over 100 mM [74].

It has previously been reported that SCFAs might have an effect on the progression of various diseases including diabetes, colorectal cancer (CRC), inflammatory bowel disease (IBD), and atherosclerosis [75,76]. Numerous studies have specifically focused on CRC [77,78]. SCFA levels were reported to have declined in CRC patients in contrast to the control group. This can be due to the reduction in the number of SCFA-synthesizing microbes which include *Roseburia* spp., *Bifidobacterium* spp., and *Lachnospiraceae* [69,79]. Furthermore, it was discovered that numerous bacterial species could synthesize tumorigenic metabolites such as secondary bile acids. On the other hand, some bacteria could produce antitumor metabolites such as SCFAs [80,81]. Significant epidemiological data in gastric and breast cancer indicated that increased rate of cancer and inflammatory diseases is induced in individuals with diets lacking SCFAs or with significantly reduced quantity of fecal SCFAs [82]. Furthermore, SCFAs from the host gut and other organs could significantly lessen cancer development, treat and inhibit lung and gastrointestinal malignancies. SCFAs do this by suppressing histone deacetylases (HDACs), thereby preventing cell growth and migration, and by activation of apoptosis [83]. Epidemiological studies have discovered that high-fiber diets are involved in low cancer incidence versus consumption of red meat which has been reported to increase cancer risk [84,85,86]. Moreover, it was discovered that SCFAs such as butyrate play a crucial role in the anticancer mechanism of high fiber diets through microbiota action [81].

### 3.1. The Mechanism of SCFAs in Cells

SCFAs display intracellular and extracellular outcomes via binding ligands to their receptors and also functioning as epigenetics modulators. SCFA receptors are found throughout the human body, and they belong to G protein-coupled receptors (GPCRs). This suggests their role in numerous cellular pathways [87]. For example, a surface receptor of macrophages, colonocytes, and adipocytes called GPR109A is frequently involved in the release of fat deposits in deprivation conditions in adipocytes [88,89]. The reduction of GPR109A expression can result in CRC pathogenesis. GPR109A receptor has been reported to be involved in T-reg-cell differentiation and establishment of proinflammatory (IL-18) and anti-inflammatory (IL-10) cytokines [88]. These responses were previously linked with cancerous outcomes, as demonstrated in Niacr1-/- mice [90]. In addition, SCFAs can also affect apoptosis and cell-cycle modulation [72]. On the other hand, activation of GPR41/43 receptor in MCF-7 cells promotes Ca^2+^ intracellular levels and stimulation of MAPK p38 [72]. These observations are widely associated with carcinogenesis and cell stress responses [91,92]. Furthermore, GPR43 receptor is absent in colon tumors and metastatic cells which suggests its role in oncogenesis. Apoptosis and G0/G1 cell cycle arrest were observed after the restoration of GPR43 expression in adenocarcinoma cell lines [93].

A potential link between SCFAs and cancer development may be due to GPCR stimulation which can subsequently activate cascades of responses resulting in cancer or prevention thereof [72]. SCFAs can also function as ligands to receptors located in the membrane, thus impacting cell metabolism. Butyrate may use sodium-coupled monocarboxylate transporter (SMCT1) during entry into the host cell. SMCT1 was initially classified as a possible tumor suppressor [72,94]. Butyrate can also utilize other carriers to propagate into the body [72]. Monocarboxylate transporter 4 (MCT4) is one of the transporters which fuses butyrate to the blood flow [95]. This enables butyrate to showcase a systemic effect on the host organism inside the bloodstream. Moreover, butyrate is able to enter back into the intestinal cavity using breast cancer resistance protein (BCRP) [72]. Decrease in BCRP mRNA expression has a potential association with colorectal adenoma pathogenesis which can possibly be linked with butyrate build-up inside the cells [96]. Post cell entry, butyrate can have an impact on histone deacetylases (HDACs) [72]. HDACs are mainly involved in cell-cycle modulation, cell proliferation, and apoptosis [97]. Furthermore, butyrate, together with its binding to HDACs, has been linked to CRC pathogenesis [97,98]. HDAC enzymatic activity becomes inhibited once butyrate binds to it. This results in gene expression modification and histone hyperacetylation. Butyrate can suppress tumorigenic cell growth via cell-cycle arrest activation and programmed cell death [72]. Lastly, butyrate might have an impact on other processes involved in epigenetic modulation, including DNA methylation, hyperacetylation of non-histone proteins, and histone methylation and phosphorylation [72,99,100].

### 3.2. The Role of SCFAs in PCa Carcinogenesis

The gut-microbiota-derived SCFAs contribute to the modulation of HDACs [101]. In return, this SCFA-mediated HDAC modulation may play a crucial role in cell homeostasis. This is because HDACs have an impact on immune-cell migration, cell adhesion, programmed cell death, chemotaxis, and cytokine synthesis. As such, manipulation of intestinal-tract SCFA levels through the alteration of the microbiota can be a potential strategy for cancer treatment and prevention. Notably, it was shown that gastric cancer and breast cancer are mediated in individuals with decreased proportions of SCFAs in feces or individuals on a diet low with SCFAs [82]. SCFAs can block cell growth, migration, HDACs, and activate apoptosis. In turn, these SCFA properties enable it to reduce cancer incidence [71]. Prior to being recognized as a microbiota-derived metabolites, SCFAs were studied as differentiation or an antiproliferative agents for treating solid tumors; for example, breast cancer and PCa [102]. Samid et al. [103] were the first researchers to study the impact of acetate on PCa. In this study, dose-dependent cell proliferation inhibition was observed after DU145 (hormone-refractory PCa cell line), LnCap (a HSPC cell line), and PC3 cell lines were exposed to phenylacetate (PA) [103]. Furthermore, tumors were not observed after PC3 cells were treated with PA and transplanted into nude mice. These results demonstrated antitumor capabilities of acetate [103]. On the other hand, Carducci et al. [104] studied the impact of butyrate on PCa development. The authors observed that the apoptotic and growth-inhibitory effects of butyrate were much higher compared to those of acetate in PCa cell lines [104]. Since these clinical studies involved a low number of participants, they were not enough to draw solid conclusions regarding acetate and butyrate effectiveness against PCa development [105,106].

Conversely, research studies have recently reported the association between SCFAs and PCa progression [101]. For example, Matsushita et al. [107] reported that SCFAs promoted PCa growth through IGF1 signaling in *Pten* knockout mice [107]. In this study, *Pten* knockout mice were utilized as a PCa model to investigate the link between animal-fat consumption and PCa pathogenesis as mediated by gut microbiota. This is because animal fat and subsequent obesity are the crucial risk factors for PCa pathogenesis, and gut microbiota composition varies with dietary composition as well as body type. Antibiotic mixture (Abx) was orally administered in these PCa mice. In addition, the mice were fed a high-fat diet (HFD) which contained high lard quantities. This resulted in a significant alteration of the gut microbiome composition, including *Clostridiales* and *Rikenellaceae* species and blocked PCa cell proliferation. It also reduced levels of circulating IGF1 and the expression of prostate *Igf1* gene. On the other hand, Abx exposure suppressed phosphatidylinositol-3 kinase (PI3K) and mitogen-activated protein kinase (MAPK) activities downstream of the IGF1 receptor in the prostate tissue. In the same study, proliferation of 22Rv1 and DU145 PCa cell lines was directly promoted by IGF1. Abx exposure also decreased levels of fecal SCFAs produced by the intestinal bacteria. On the other hand, SCFA supplementation promoted tumor growth through the increasing of IGF1 levels. Notably, IGF1 was reported to be highly expressed in PCa tissue from patients with obesity. The authors concluded that IGF1 synthesis, as stimulated by gut-microbiota-derived SCFAs, promotes PCa by activating localized PI3K and MAPK signaling pathways (Figure 3) [107].

In another study, Matsushita et al. [67] observed a relative increase in bacterial strains producing SCFAs in high-grade PCa. These strains were *Lachnospira*, *Rikenellaceae*, and *Alisipes*. Furthermore, the authors also observed an increase in *Lachnobacterium*, *Subdoligranulum*, and *Christensenellaceae* in high-grade PCa. These findings suggest that SCFAs may play a role in PCa progression [67]. Liu et al. [108] recently transplanted fecal suspension from a castration-refractory prostate cancer (CRPCa) patient into a transgenic adenocarcinoma of the mouse prostate (TRAMP) mouse model. This transplantation resulted in the acceleration of cancer progression in the TRAMP mouse. In this study, SCFAs increased in vitro migration and invasion of PCa cells [108].

### 3.3. The Effect of Microbiome-Derived SCFAs on Response to Cancer Immunotherapy

State-of-the-art techniques have shown that microbiota influence carcinogenesis and immunotherapy [109]. Moreover, microbiota can positively or negatively impact tumorigenesis [110]. Bacteria synthesize cancerous compounds or toxins which can result in inflammatory or immunosuppressive responses that support carcinogenesis [109]. Conversely, gut microbiota may inhibit oncogenesis by boosting antitumor immunity [110]. In addition, gut microbiota impair anticancer treatment and toxicity efficacy by changing systemic and local immune responses [111]. Analysis of microbial-derived metabolites such as SCFAs, as well as gut microbiota imbalance with regard to their effects on immune responses, will ameliorate understanding of numerous frequent etiological disorders [112]. SCFAs, particularly propionate, butyrate, and acetate, are found in certain concentrations. Furthermore, SCFA amounts can change with age, disease, and diet. SCFA concentrations are particularly regulated by the gut microbiota proportions. In addition, gut dysbiosis can induce imbalance of the synthesized SCFAs. It was also observed that SCFAs can inhibit HDAC activity which is involved in deacetylation and histone crotonylation. These SCFA attributes can support pro/anti-inflammatory hemostasis and potentiate their immunomodulatory capabilities. SCFAs influence gut immune cells and modulate the immune system through multiprotein inflammasome complexes. They also have localized functions in the intestines which are occupied by gut bacteria [112]. Furthermore, SCFAs are important for immune regulation [113]. Butyrate has systemic anti-inflammatory properties through alteration of cytokine expression of immune cells and having an impact on cellular processes such as activation, propagation, and apoptosis. Butyrate particularly inhibits HDAC. On the other hand, HDAC prohibits gene transcription by retaining the compact structure of the chromatin. Therefore, inhibition of HDAC by butyrate leads to hyperacetylation. In this way, butyrate can regulate gene expression and exert its antiproliferative properties. (Figure 4) [114].

Butyrate is commonly linked with immune modulation, particularly via activation of T-reg cells [115]. Research studies have reported that the human intestinal microbiota structure is related to antitumor efficacy in metastatic melanoma patients exposed to anti- cytotoxic T-lymphocyte-associated protein 4 (CTLA-4) and anti-programmed cell death protein 1 (PD-1) monoclonal antibodies [116,117,118]. Furthermore, composition of the gut microbiota seems to be linked with increased risk of anti-CTLA-4-stimulated colitis in patients with metastatic melanoma [118,119]. These findings suggest the link between the presence of specific bacteria and toxicities and/or clinical response in metastatic melanoma patients. Notably, elevated proportions of *Faecalibacterium* and other *Firmicutes* have been associated with good clinical response to ipilimumab/anti-PD-1, anti-PD-1, and ipilimumab treatments [118,120]. Based on these findings, *Faecalibacterium* might be significantly mediated to clinical responses in metastatic melanoma patients who were exposed to immune checkpoint inhibitors [121].

Regarding studies focusing on the link between the gut microbiome structure and clinical reaction, as well as the effect of SCFAs on the immune responses, Coutzac et al. [121] studied the activation of anticancer reaction due to anti-CTLA-4 and systemic bacterial-derived SCFA inhibition. The authors showed systemic bacterial-derived SCFAs modulate anti-CTLA-4-stimulated immune responses as well as their anti-tumor effectivity. This study showed that systemic propionate and butyrate limit anti-tumor attributes of anti CTLA-4 [121]. These findings indicate a link between the human gut bacterial structure and the clinical responses to ipilimumab via microbiota-derived metabolites. As these microbiome-derived SCFAs play different roles in the immune system in various outcomes, they can be involved in immunotherapy for treating intestinal and extraintestinal-tract malignancies. As such, more investigations are needed to establish specific roles and mechanisms of various bacterial species and their metabolites to ameliorate cancer treatment, especially PCa since not much is known about the effects of these metabolites on its progression and treatment [71].

### 3.4. Gut Microbiota and SCFA Profile Disparities among Different Race/Ethnic Groups

Several studies have reported racial/ethnic disparities in terms of bacterial content in the gut microbiome and SCFA composition [122]. These studies explored microbiome and SCFA racial disparities in various cancers, including CRC, colon cancer, cervical cancer (CC), breast cancer, and endometrial cancer (EC). Notably, there are currently no published studies exploring racial/ethnic disparities in microbiome/SCFA composition pertaining to PCa carcinogenesis. Hester et al. [123] compared bacterial and SCFA content from 20 stool samples of healthy participants. The study cohort comprised five groups of Non-Hispanic Whites (NHWs), Non-Hispanic Blacks (NHBs), American Indians, and Hispanics. The authors reported lower levels of butyrate, acetate, and total SCFA content, and a higher pH in NHBs versus other three racial groups. Furthermore, they reported higher levels of *Firmicutes* in NHBs compared to Hispanics and Whites. However, the authors observed lower levels of *Lachnospiraceae*—a bacteria involved in butyrate synthesis. Additionally, the authors observed a higher ratio of *Firmicutes* compared to *Bacteriodes* among Blacks [123]. This ratio has previously been implicated in obesity [122]. In another study, Farhana et al. [124] reported higher levels of *Bacteroides* in colonic effluents of African Americans (AAs) versus NHWs, particularly *Enterobacter* and *Fusobacterium nucleatuma* species. On the other hand, *Bifidobacterium* and *Akkermansia muciniphilia* levels were higher in NHWs compared to AAs. In the same study, the authors observed a decrease in microbial diversity in AAs versus NHWs [124]. Carson et al. [125] reported an abundance of *Bacteroides* in AAs versus NHWs in a study investigating the link between psychological stress and gut microbiome in a healthy cohort of Black and White women. Mai et al. [126] investigated whether variations in dietary habits between NHBs and NHWs were linked with genotoxicity/cytotoxicity of fecal water and fecal microbiota composition [126]. The authors reported higher *Bacteroides* stool levels in NHBs compared to NHWs. They also observed lower dietary magnesium and calcium, as well as lower levels of vitamins A, C, D, and E [126]. Yazici et al. [127] compared and contrasted colonic biopsies of healthy mucosa from CRC patients and healthy controls. The authors reported a greater sulfidogenic abundance in NHBs versus NHWs between cases and controls. Furthermore, *Pyramidobacter* and *Bilophilia wadsworthia* spp. were markedly higher in AA cases versus the healthy controls. Notably, the authors observed an increased meat, fat, and protein intake in NHBs [127]. David et al. [128] reported that diet can modify the human gut microbiome. In another study, gut microbiota were reported to be central in modulation of diet-induced obesity in lymphotoxin-deficient mice [129]. Obesity is commonly known as the risk factor in various cancers [122].

Comparing the gut microbiota among different populations with vast dietary variation will provide more information on how the gut microbiota and diet can induce carcinogenesis [122]. In this regard, Ou et al. [130] investigated variations between gut microbiomes of NHBs and native Africans (NAs). The authors observed higher SFCAs, total bacteria, and microbial genes in NAs, and a higher secondary bile acid in fecal samples of NHBs. These observations suggest a lower proteolytic fermentation and a higher saccharolytic fermentation in NAs versus NHBs [130]. Regarding specific bacterial species in this study, *Bacteroides* were more common in NHBs, whilst *Prevotella* was more abundant in NAs. The authors attributed these observations to the fact that NHBs consume more fat and dietary meat, but less fiber and complex carbohydrates [130]. In another study by the same authors, levels of potentially cancerous secondary bile acids, namely deoxycholic and lithocholic acids were higher in NHWs and NHBs versus NAs. This was a study whereby a small sample for high colon cancer risk was compared among the three race groups [131]. Nava et al. [132] also showed that NAs harbored a more diverse methanogenic Archaea population in contrast to NHWs and NHBs. Furthermore, populations of other hydrogenotrophic bacteria such as sulphate-reducing bacteria were also more defined in NAs [132]. O’Keefe et al. [133] examined total colonic evacuants for minerals, nitrogen, calcium, and SCFAs and discovered that total butyrate and SCFA levels were higher in NAs versus NHBs. However, zinc, calcium, and iron were lower in NAs versus NHWs and NHBs. These findings support the mediatory role of these chemicals in terms of the impact of microbiota on colon cancer pathogenesis [133]. NAs were previously reported to have a propensity for methanogenic instead of sulfidogenic disposal of hydrogen synthesized from microbial fermentation in the human colon [122]. In another study, O’Keefe et al. [134] carried out a study to determine factors that contribute to low colon-cancer cases observed among Black versus White South Africans. The authors reported that NAs released more methane in their breadth compared to their White counterparts. In addition, stool samples from populations with European lineage showed more sulphate-reduction activity. Variations in the bacterial composition of the stool samples in NHBs and NAs are due to differences in diet [134]. Notably, Africans consume more coarse vegetables and grains [122].

Recently, Vikramdeo et al. [135] examined resident microbial compositions in the cervical intraepithelial lesions of AA, Hispanic, and NHW women who were previously screened for CC risk assessment. The authors observed a significant decrease in beneficial *Lactobacillus* abundance in the CIN lesions of Hispanic and AA versus NHW women. Variable abundance of potentially carcinogenic *Gardnerella*, *Prevotella*, *Fastidiosipila*, and *Delftia* was also observed between the three racial groups. In addition, increased Micrococcus composition was also observed in HIS and AA compared to NHW women. The authors concluded that microbial dysbiosis in the cervical epithelium due to an increase in the ratio of pathogenic to beneficial microbes might be linked with increased risk disparities in CC [135]. Smith et al. [136] investigated microbiota from breast cancer and healthy tissues from NHB and NHW women to determine unique microbial signatures by race, tumor subtype, or stage. Notably, breast cancer tissues obtained from NHB women had an increased *Ralstonia* abundance versus those from NHW women which could potentially explain racial disparities in breast cancer. In addition, tumor subtype analysis showed abundance of *Streptococcaceae* specifically in triple-negative breast cancer (TNBC). The authors noted that this was the first research study to determine racial disparities in the breast tissue microbiota between NHW and NHB women [136]. In the following year, Thyagarajan et al. [137] conducted 16S RNA gene-based sequencing on retrospective tumor samples and compared them to healthy tissue samples obtained from NHB and NHW women. The analysis of the tissue samples for microbiota composition showed significant differences in the abundance of specific taxa at genus and phylum levels between BNH and WNH women [137]. Bridges et al. [138] carried out a study aimed at exploring how colonoscopy results link with stool SCFA levels, inflammation markers, and dietary intake in a diverse group of average-risk adults. Study participants were scheduled for routine screening colonoscopies for colorectal cancer. The authors reported that AAs had higher total SCFA levels in the stool samples versus other racial groups. In addition, AAs had a markedly lower intake of non-starchy vegetables and similar colonoscopy outcomes as well as inflammatory marker expression versus other racial groups [138].

Hawkins et al. [139] analyzed 95 early-stage ECs between AA and NHW women. The authors reported an increase in microbial diversity as well as high abundance of *Cyanobacteria*, *Firmicutes*, and *OD1* phyla in AA women compared to their NHWA counterparts. Furthermore, abundance levels of *Geobacillus* and *Dietzia* were lower in tumors obtained from AA versus NHW women. Comparison of ECs between overweight AA and overweight NHW women revealed five bacterial distributions. A higher *Lactobacillus acidophilus* abundance was observed in ECs obtained from AA women. The authors concluded that an increase in microbial diversity together with distinct microbial profiles between overweight AA and overweight NHW women suggests that intra-tumoral bacterial species might be central to the observed racial disparities and carcinogenesis [139]. Lastly, Carson et al. [140] recently conducted a case-control study comparing women who were newly diagnosed with CRC matched with race, age, and body mass index (BMI) in a 1:2 ratio. NHW women with CRC showed increased abundance of *Gemellaceae*, *Peptostreptococcus*, and *Fusobacteria* versus other race–cancer combination groups. These findings suggest that the links between microbiome and CRC may vary by race/ethnicity [140]. Although there is lack of scientific data regarding racial/ethnic microbiome and gut microbiota and SCFA profile disparities in PCa, there is substantial evidence of these disparities in other cancers. Therefore, extensive research is needed to establish these disparities in PCa, as this will assist with understanding underlying mechanisms in PCa carcinogenesis. In return, this will help in developing novel and efficacious PCa treatment strategies tailored for each patient or even ameliorate current treatment strategies.

### 3.5. The Effect of Microbiome-Derived SCFAs on PCa Progression via Epigenetics

Gene expression is epigenetically regulated at three main levels, namely chromatin modification, non-coding RNAs, and DNA methylation [141]. DNA methylation is central in gene silencing and can also modify the chromatin architecture [142,143]. There is an abundance of CG-rich sequences known as CpG islands in the transcription start site of genes. These CpG islands are unmethylated in normal cells. However, their methylation results in silencing of the corresponding gene. DNA is wrapped around proteins called histones which undergo different posttranslational modifications (PTMs). Histone PTMs modulate architecture of the chromatin, and certain PTMs are linked with different transcriptional processes that can result in silencing or activation of specific genes. For example, histone methylation, sumoylation, deamination, and proline isomerization correlate with gene silencing. Conversely, histone methylation, acetylation, ubiquitylation, and phosphorylation are associated with gene activation [144]. Gene expression can also be regulated through regulatory and non-coding RNAs [145]. The roles of microRNA (miRNA) and long no-coding RNA (lnRNA) in gene regulation have previously been elucidated [146,147,148,149]. On the other hand, other regulatory RNAs such as enhancer RNA, small nucleolar, and nuclear RNA regulate and influence different transcriptional processes. Furthermore, their roles in maintenance of homeostasis have started to emerge [150,151,152]. Since epigenetic modifications are heritable, it is critically important to understand factors such as microbial dysbiosis which influence transcriptional homeostasis and epigenetic memory. This will help in establishing underlying molecular mechanisms in various human disorders, cancer included [153,154].

Molecular mechanisms underlying microbiome-induced host epigenetic modifications are not yet fully understood [141]. However, microbiome-derived metabolites such as butyrate, biotin, and folate have the potential to induce epigenetic modifications [155]. The effect of three main microbiome-derived SCFAs on chromatin structure regulation are summarized in the Table 1 below. However, it is important to note that there are currently no published research studies exploring the effect of microbiome-derived SCFAs on epigenetic modifications in PCa. Therefore, more in-depth mechanistical studies are needed to establish whether SCFAs have an impact on epigenetic changes leading to PCa progression.

## 4. Conclusions and Future Perspectives

The importance of the human microbiome in disease and health is an important topic in scientific research. Although numerous studies have been published regarding the microbiome and PCa tumorigenesis, it is still not clear whether the human microbe is a contributory or causative agent to PCa [163]. However, current data indicate that the human microbiome plays a crucial role in PCa. With that being said, the exact mechanism of how the microbiome contributes to PCa development at a molecular level is yet to be fully unraveled [163]. As previously mentioned, the microbiome can influence systemic and local immune responses using numerous metabolites including SCFAs [111]. Despite lack of scientific data in terms of the effect of SCFAs on PCa pathogenesis, studies on other malignancies suggest that SCFAs may have a significant impact on PCa progression [101]. Therefore, further studies need to be conducted to investigate mediation of SCFAs in PCa to determine their exact roles in PCa pathogenesis and treatment. In addition, further investigations are needed to evaluate SCFA-mediated immune-regulatory pathways which will help determine clinically actionable targets for precision medicine in PCa. Numerous studies have reported racial/ethnic disparities in terms of gut microbiome composition and SCFA content in various cancers, as previously mentioned [122]. However, these disparities have not yet been reported in PCa carcinogenesis. Therefore, this highlights the urgent need for more research to establish the existence of these racial/ethnic disparities and underlying mechanisms in PCa carcinogenesis. The effects of microbiome-derived SCFAs on host epigenetic modifications in PCa have not yet been fully explored. Therefore, more research is warranted in terms of establishing how these metabolites contribute to PCa progression via epigenetic modifications. Together, this will help in personalized PCa incidence prediction, prognosis, identification of potential microbial targets, and development of new treatment and behavioral strategies.

## Figures and Tables

**Figure 1 cancers-15-04086-f001:**
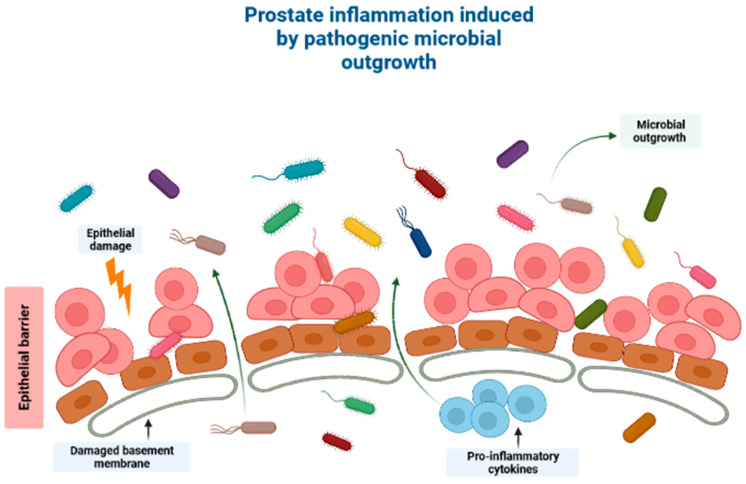
A schematic diagram showing how the human microbiome induces cancer pathogenesis, specifically through induction of prostatic inflammation. Outgrowth of pathogenic microorganisms might take place in the luminal portion of the prostate ducts under pathological conditions. This outgrowth might be due to one of several reasons, including introduction of the microorganisms into the prostate organ through urine reflux, dysbiosis or pathogenic outgrowth of the microbiota in the prostatic urethra, changes to the antimicrobial part of the prostatic fluid due to prostate atrophy, or a combination of the processes mentioned above. Direct damage to the epithelial barrier via urine reflux, for example, can allow microorganisms to invade the epithelial cells, and thereby induce prostate inflammation. Created with BioRender.com (Accessed on 31 July 2023).

**Figure 2 cancers-15-04086-f002:**
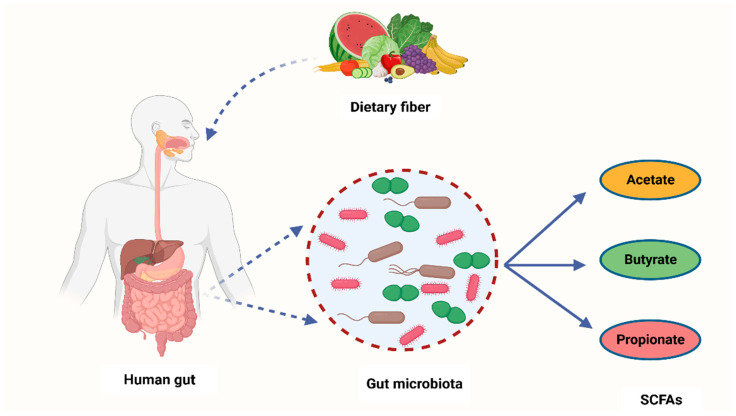
A schematic diagram showing how bacteria from the human microbiome convert dietary fiber into three main SCFAs—acetate, butyrate, and propionate. Acetate is synthesized through conversion of pyruvate. This occurs in three ways: either through the reductive acetyl-CoA pathway, the Wood–Ljungdahl pathway, or directly through acetyl-CoA. On the other hand, butyrate is a product of acetyl-CoA reduction to butyryl CoA, which subsequently becomes converted into butyrate-by-butyrate kinase and transbutyrylase enzymes. Moreover, butyryl CoA may form butyrate through butyryl-CoA transferase-acetate. Lastly, propionate is formed through the succinate pathway. It can also be synthesized via the acrylate pathway from lactate (precursor) with hexoses and pentoses (simple sugars) acting as reaction substrates. Propionate can also be formed through the propanediol pathway whereby rhamnose and fucose (deoxyhexoses) function as substrates. SCFAs—short-chain fatty acids, Acetyl-CoA—acetyl coenzyme A. Created with BioRender.com (Accessed on 4 July 2023).

**Figure 3 cancers-15-04086-f003:**
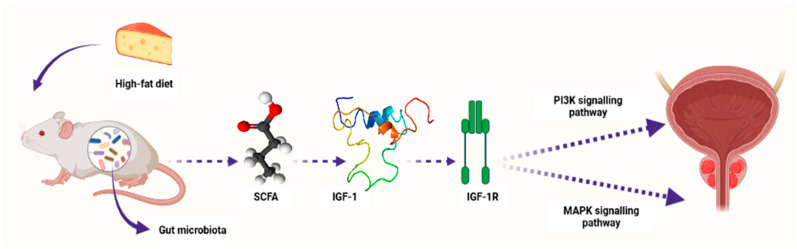
A schematic diagram showing how microbiota-derived SCFAs induced PCa carcinogenesis by activating the PI3K and MAPK signaling pathways through IGF1 and its receptor IGF-1R in *Pten* knockout PCa mice. Composition of the gut microbiota is influenced by diet content, e.g., high-fat diet. SCFAs—short-chain fatty acids, IGF-1—insulin-like growth factor-1, IGF-1R—insulin-like growth factor-1 receptor, PI3K—phosphatidylinositol-3 kinase, MAPK—mitogen-activated protein kinase. Created with BioRender.com (Accessed on 31 July 2023).

**Figure 4 cancers-15-04086-f004:**
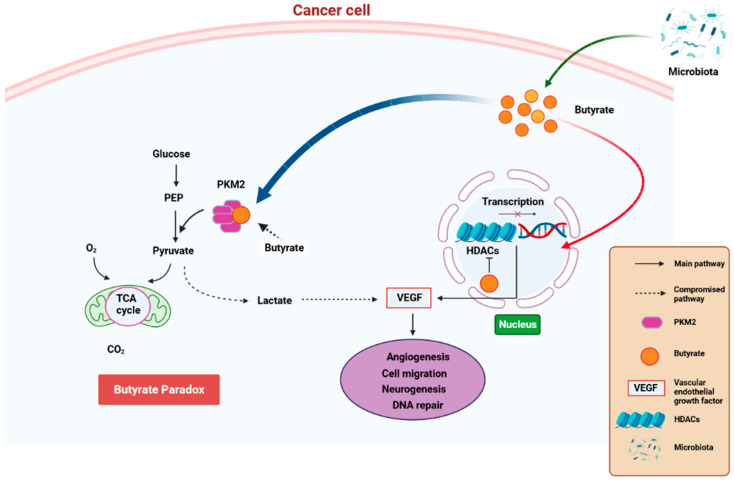
A schematic diagram showing how butyrate exerts anti-cancer properties. Butyrate is synthesized is a by-product of fiber fermentation by microbiota and then accumulates in cancerous gastric epithelia through the Warburg effect. This result in glucose metabolism and increased lactate synthesis by the epithelial cells. In return, lactate induces processes that trigger the upregulation of VEGF, resulting in the upregulation of angiogenesis, cell migration neurogenesis, and aberrant DNA repair. These processes result in cancer pathogenesis. Butyrate effectivity against cancer occurs through the butyrate paradox in two ways: (1) butyrate travels from the cytoplasm to the nucleus where it acts as an HDAC inhibitor by terminating cell-cycle continuation via altered gene expression and (2) butyrate can reverse metabolism anaerobic glycolysis to OXPHOS by binding to PKM2. This, in return, renders PKM2 a more active dephosphorylated tetrameric version, thus favoring energy creation via the Krebs cycle. HDAC—histone deacetylase, OXPHOS—oxidative phosphorylation, PKM2—pyruvate kinase M2. Created with BioRender.com (Accessed on 25 June 2023).

**Table 1 cancers-15-04086-t001:** A summary of the effect of three main microbiome-derived SCFAs (acetate, butyrate, and propionate) on chromatin structure regulation. Adapted from [155].

**Microbiome** **derived-SCFAs**	**Metabolite**	**Effect on Host Epigenetic Changes: Putative or Demonstrated**	**Model System**	**Outcome**
	Acetate	Putative	In vitro, mammalian cell culture	Increases histone acetylation [156]
	Butyrate	Demonstrated	In vitro, mammalian cell culture, human, mouse	HDAC inhibition [157,158], HAT activation, and protection from colorectal cancer [159,160]; protection from HFD-induced metabolic syndrome, and is linked with deceased HDAC activity [161]; increase in histone acetylation in HT-29 cells [98]; modest increase in *HDAC3* and *HDAC5* expression in gut organoids [162]
	Propionate	Putative	In vitro mammalian cell culture; intestinal organoids	Weak–modest HDC inhibition in vitro (<butyrate) [158]; increase in histone acetylation in HT-29 cells [98]; modest increase in *HDAC3* and *HDAC5* expression in gut organoids [162]

SCFAs—short-chain fatty acids, HDAC—histone deacetylase, HAT—histone acetyltransferases, HFD—high-fat diet.

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
