# Peer review of "Dissecting Microbiome-Derived SCFAs in Prostate Cancer: Analyzing Gut Microbiota, Racial Disparities, and Epigenetic Mechanisms"

_cancers, 2023, doi:10.3390/cancers15164086_

Round 1
Reviewer 1 Report
Dear Authors,
Cancers-2540788
Dissecting Microbiome-Derived SCFAs in Prostate Cancer: Analyzing Gut Microbiota, Racial Disparities, and Epigenetic Mechanisms by Miya et al is an interesting article. The authors are summarizing the racial disparities in terms of 15 microbiome composition and SCFAs content across different human cancers, including colon, cervical, and colorectal cancers. Also, the authors are suggesting that epigenetic modifications also play a 17 crucial in carcinogenesis and response to various treatment interventions. However, before making a positive decision the following are need to be addresses.
Comments:
1. Association between the human microbiome and PCa pathogenesis. Under this title Fig 1 show the presence of microbiome in male. The figure and title are not matching. The figure is not describing the mechanism of male microbiome induced oncogenesis of PCa.
2. The authors state that male and female microbiome are different. in general microbiome in the human body depends on the food and environment, moreover in a family or a small community consuming the same food and exposed the same environment. How do you explain this situation in terms of microbiome.
3. Fig 2 describes SCFAs induced PCa. Please provide the detailed mechanism of SCFAs induced/associate/response for PCa development with signaling pathways
4. Microbiome-derived short chain fatty acids (SCFAs). Fig 3 lacking positive and negative effects of acetate, butyrate, and propionate.
5. In the simple summary the authors claim that SCFAs promotes PCa (line 14), how come that butyrate exerts anti-cancer activities in Fig 4. This statement is confusing.
Minor edit required
Author Response
Point 1: Association between the human microbiome and PCa pathogenesis. Under this title Fig 1 show the presence of microbiome in male. The figure and title are not matching. The figure is not describing the mechanism of male microbiome induced oncogenesis of PCa.
Response 1: Fig 1 has been removed since it is not completely detailed. However, a new figure has been added but under in the introduction section.
Point 2: The authors state that male and female microbiome are different. in general microbiome in the human body depends on the food and environment, moreover in a family or a small community consuming the same food and exposed the same environment. How do you explain this situation in terms of microbiome.
Response 2: The microbiome composition between males and females is specifically different in the urogenital tract. This is due to hormonal differences and the anatomical structure between age and gender. The gut microbiome composition is the one that is specifically determined by several factors, including diet, body type etc.
Point 3: Fig 2 describes SCFAs induced PCa. Please provide the detailed mechanism of SCFAs induced/associate/response for PCa development with signalling pathways.
Response 3: In the initial manuscript, the association between SCFAs and PCa were generalized to humans. However, the detailed mechanism has been included in the revised manuscript, and it is directly from the primary reference according to what the authors reported. In that particular study, the mechanism is in relation to Pten knockout PCa mice. Therefore, the exact mechanism of this association in humans is yet to be fully elucidated since this a relatively new field of research in oncology. The new diagram has been moved to Section 3.2.
Point 4: Microbiome-derived short chain fatty acids (SCFAs). Fig 3 lacking positive and negative effects of acetate, butyrate, and propionate.
Response 4: SCFAs have been shown to specifically promote PCa carcinogenesis. However, more research still need to be carried out to determine the underlying molecular processes. Nevertheless, Butyrate has specifically been reported to have antitumour properties in colorectal cancer as discussed below.
Point 5: In the simple summary the authors claim that SCFAs promotes PCa (line 14), how come that butyrate exerts anti-cancer activities in Fig 4. This statement is confusing.
Response 5: It is important to note that this area of oncological research is still relatively new and therefore, a lot of factors are still not yet known in terms of the relationship between SCFAs and cancer, especially in PCa. The purpose of the Fig 4. Is to demonstrates how butyrate can fight off cancer through the ‘Butyrate Paradox’ process. I have revised the diagram and also expanded the figure legend to make explain the effect of butyrate on cancer in more details. However, not much is known with regard to the effects of butyrate in PCa.
Reviewer 2 Report
In this article, the authors review the topic of microbiome-derived short chain fatty acids (SCFAs) in prostate cancer.
General comment:
The topic is relevant and important.
Specific points:
Figure 1 does not seem quite informative; is there a way to improve it with more information? This is only a suggestion and I leave it up to the authors to decide on its suitability.
Why is section 4 singled out? Would it be better to fit it before section 3.3. or 3.4.?
Minor points:
When there are more references cited, they should be written together, e.g. [13-17] (line 77) and not separately.
Line 174: 'Prevotella' is listed twice.
Figure 4: It would be good to write on the Figure which shape (orange circle) is butyrate.
Line 435: It is written 'ethic' instead of 'ethnic'.
Minor editing of English language required.
Author Response
Point 1: Figure 1 does not seem quite informative; is there a way to improve it with more information? This is only a suggestion and I leave it up to the authors to decide on its suitability.
Response 1: Figure 1 has been removed and a new figure has been included.
Point 2: Why is section 4 singled out? Would it be better to fit it before section 3.3. or 3.4.?
Response 2: Section 4 has been moved and numbered as section 3.4.
Point 3: When there are more references cited, they should be written together, e.g. [13-17] (line 77) and not separately.
Response 3: The references have now been correctly cited.
Point 4: Line 174: 'Prevotella' is listed twice.
Response 4: Line 174 has been corrected.
Point 5: Figure 4: It would be good to write on the Figure which shape (orange circle) is butyrate.
Response 5: Figure 4 has been revised and components annotated accordingly.
Point 6: Line 435: It is written 'ethic' instead of 'ethnic'.
Response 6: Line 435 has been corrected.
Round 2
Reviewer 1 Report
Manuscript quality is improved